# Stress and Right Prefrontal Transcranial Direct Current Stimulation (tDCS) Interactive Effects on Visual Working Memory and Learning

**DOI:** 10.3390/brainsci13121642

**Published:** 2023-11-27

**Authors:** Yael L. E. Ankri, Yoram C. Braw, Oded Meiron

**Affiliations:** 1Department of Psychology, Ariel University, Ariel 4077625, Israel; yael.errera@mail.huji.ac.il (Y.L.E.A.); yoramb@ariel.ac.il (Y.C.B.); 2Faculty of Education, Bar-Ilan University, Ramat-Gan 5290002, Israel; 3Clinical Research Center for Brain Sciences, Herzog Medical Center, P.O. Box 3900, Jerusalem 9103702, Israel

**Keywords:** transcranial direct current stimulation (tDCS), dorsolateral prefrontal cortex (dlPFC), social stress, trier social stress test (TSST), *n*-back task

## Abstract

Stress impacts prefrontal cortex (PFC) activity and modulates working memory performance. In a recent study, stimulating the dorsolateral PFC (dlPFC) using transcranial direct current stimulation (tDCS) interacted with social stress in modulating participants’ working memory. More specifically, stress disrupted the enhancing effects of dlPFC tDCS on working memory performance. The current study aimed to further explore these initial findings by randomizing healthy females to four experimental conditions (*N* = 130); stimulation (right dlPFC tDCS vs. sham) and *stress manipulation* (social stress vs. control). Participants performed cognitive tasks (i.e., visual working memory task and a visual declarative memory task) at baseline and post-stimulation. They also completed self-report measures of stress and anxiety. A significant *stimulation* × *stress* interaction was evident in the declarative memory (One-Card Learning, OCL) task, while working memory performance was unaffected. Though tDCS stimulation and stress did not interact to affect working memory, further research is warranted as these initial findings suggest that immediate visual-memory learning may be affected by these factors. The limited number of earlier studies, as well as the variability in their designs, provides additional impetus for studying the interactive effects of stress and tDCS on human visual learning.

## 1. Introduction

Transcranial direct current stimulation (tDCS) is an accessible, safe, and inexpensive neuromodulation technique [1,2,3]. It has attracted growing research attention as a non-invasive intervention that improves the cognitive functioning scores of clinical patients and healthy individuals [4,5,6]. Initial studies suggested enhanced working memory performance after dorsolateral prefrontal cortex (dlPFC) tDCS stimulation (see review [7]). However, the findings of later studies were mixed and significant effects, when found, tended to be small [8,9]. Importantly, these initial findings point toward exposure to stress as an important, though understudied, factor that moderates the effect of tDCS on specific cognitive functions, particularly working memory [10,11,12].

We recently examined the effects of anodal tDCS over the right dlPFC on working memory performance [13]. The dlPFC was chosen as it represents a critical anterior frontal neural network hub that is essential for facilitating immediate access to target stimuli representations and the mediation of action contingencies underlying the temporal organization of behaviors. More so, selective attention processes and the retention of relevant visual–spatial working memory items can be improved by employing excitatory repetitive TMS (rTMS) over the left dlPFC, suggesting that dlPFC rTMS could also improve athletic visual–motor performance, which usually demands fast visual–spatial orienting attention responses in dynamic “team-sport” environments [14]. Based on this accumulated knowledge, we assessed the effects of stimulation (active vs. sham) on verbal working memory performance were tested in two conditions (exposure to social stress vs. control condition) in our earlier study. As expected, the stress manipulation increased both objective and subjective indicators of stress (i.e., cortisol levels and self-report questionnaires). Notably, stress significantly interacted with the tDCS stimulation, suggesting that applying right anodal dlPFC tDCS *or* inducing moderate stress enhances working memory performance. Reaching an optimal zone of performance, mediated by more efficient PFC regulation, was tentatively suggested to underly these effects. In contrast, combining tDCS stimulation and stress seemed to impair performance, perhaps due to the increased involvement of PFC regulatory circuits and compromised top-down control of ongoing working memory processes. However, the post hoc analyses exploring the source of the interaction between tDCS stimulation and stress were not significant [13].

The current study explored the possible interaction between right anodal dlPFC tDCS stimulation and stress using a randomized controlled trial that was performed on a cohort of healthy young females. Importantly, we incorporated insights from our earlier investigation. First, we enhanced statistical power by increasing the sample size. Second, the verbal working memory task utilized by Ankri, Braw [13] was developed to study executive attention deficits in neuropsychiatric patients [15,16]. Consequently, a ceiling effect may have masked genuine effects in healthy participants (see also [17]). Therefore, in the current study, we employed a working memory task (*2*-Back) derived from the Cogstate computerized battery [18,19]. The *2*-Back requires the rapid sensory integration of visual stimuli and continuous updating of target items in visual working memory and, importantly, is likely more sensitive than the task used previously by Ankri, Braw [13]. Third, the current study assessed baseline intraindividual differences in cognitive functioning, while our earlier study evaluated cognition only during post-stimulation assessments. Finally, we expanded our cognitive testing procedures beyond verbal working memory mechanisms by testing incidental visual learning. More specifically, we included a second Cogstate task (One-Card Learning task, OCL), which is considered a sensitive measure of visual recognition learning [20,21,22]. Both the *2*-Back and OCL tasks employ the same visual-stimulus modality (i.e., playing cards). However, they are based on two different functional networks; while the *2*-Back task assesses visual working memory and is closely related to dlPFC functioning, the OCL task assesses incidental visual learning and is more reliant on cortical-hippocampal functioning [23]. Both visual working memory and visual learning neural networks share regulatory PFC mechanisms [24] responsible for maintaining the storage of information in working memory and long-term memory, respectively.

Considering the experimental modifications that were made in comparison to the original experimental protocol [13], we expected to replicate our earlier findings and provide more substantial empirical support for the interaction between right-prefrontal tDCS and stress on visual working memory functioning. Right-hemispheric processing seems to facilitate the visual learning of new visual features [25]. Therefore, we hypothesized that right-prefrontal excitatory stimulation and stress would also impact visual learning. Considering the pronounced reliance of working memory on *bilateral* prefrontal function [24,26], we expected the interaction of right-prefrontal tDCS and stress to impact working memory to a lesser degree than its impact on visual memory.

## 2. Method

### 2.1. Participants

Healthy female adults were assessed for eligibility (*N* = 130). Participants were undergraduate students who received course credit for participating in the study. Inclusion criteria were as follows: aged 18–40, right-handed, and native Hebrew speakers with normal or corrected-to-normal vision. Exclusion criteria were as follows: (a) Self-reported past or present major neuropsychiatric, developmental, or substance use disorders. (b) Any contraindications for performing tDCS (e.g., history of head injury, chronic dermatological disease, and pregnancy). See Figure 1 for CONSORT diagram and Table 1 for participants’ demographic and baseline data. In the current study, we aimed to increase the groups relative to our previous study ) and relative to what is customary in the field. Regarding the latter, the average sample size is 14.6 participants for between-subjects designs and 17.9 for within-subject designs in brain stimulation studies [27]. We also used G*POWER 3.1.9.2 [28,29] to determine the sample size needed to detect an effect if one exists. Based on Mancuso, Ilieva [9], 28 participants per condition were deemed adequate, even assuming a small effect size. Note, also, that due to gender differences in lateralization of prefrontal network activation during verbal working memory storage [30], we recruited only female participants. This was expected to to increase the homogeneity of the sample and, thereby, further decrease variance in outcome measures and increase statistical power.

### 2.2. Tools

#### 2.2.1. Cogstate Battery

Cogstate is a computerized battery that was used to study diverse neuro-psychiatric disorders e.g., [31,32,33]. All Cogstate tasks utilize similar visual stimuli and general design; playing cards appear one at a time, face down, on the computer screen (inter-stimulus interval, ISI, ranges between 500 and 1500 ms). The participant then presses a pre-defined keyboard button (representing ‘yes’/‘no’) as quickly as possible, depending on the specific instructions of the task. A small beep is sounded when they press the wrong key. The following tasks were used in the current study: (a) *2-Back*: Visual 2-back working memory task in which the participant determines if the card that they are shown is identical to the one shown two cards previously (duration = 4 min). (b) *One-Card Learning (OCL)*: Visual memory and learning task in which the participants determine whether the current card was presented earlier in the task (duration = 6 min). See additional information at https://www.cogstate.com/ (accessed on 3 November 2023).

#### 2.2.2. Stimulation Parameters

Stimulation was delivered by a battery-driven constant current stimulator (Chattanooga Ionto, Iontophoresis System, Hixson, TN, USA) using a pair of saline-soaked synthetic sponge electrodes. The electrode size was 5 × 5 cm. Following Meiron and Lavidor [30], the electrode montage was Cz for the reference cathode (according to the 10–20 international system for EEG electrode placement) and the right dlPFC for the anode (corresponding to F4/AF4 in the 10–20 system). The stimulation site was located following Fitzgerald, Maller [34]. When delivering active tDCS, the current was applied for 20 min with a fade-in/fade-out ramp of 30 s. The current intensity was 2.0 mA. The same fade-in/fade-out ramp was used when delivering sham stimulation, but the constant current lasted only 30 s. All participants were informed that they were receiving active stimulation (single-blind paradigm).

### 2.3. Procedure

Participants were randomly assigned to one of the experimental conditions. They then filled out the State-Trait Anxiety Inventory (STAI), a commonly used 20-item anxiety inventory [35,36], and a visual analog scale (VAS), which measurs subjective stress on a scale of 0 to 100, as used in [37]. Next, they performed the two Cogstate tasks (baseline cognitive assessment). After taking head measurements to secure electrode placement according to the planned montage, participants received 20 min of either active or sham unilateral stimulation to the right dlPFC. Next, participants underwent one of two stress manipulations: (a) *Trier Social Stress Test (TSST)*: well-validated stress manipulation [38,39], which simulates a job interview and includes three stages (each lasting 5 min). While Ankri, Braw [13] used a modified protocol (i.e., the mental arithmetic test, part of the TSST, was not performed; pp. 106–107), the original TSST procedure was used in the current study. (b) *Friendly-TSST (control)*: this condition, termed noStress in the context of the current study, preserves components of the TSST while reducing its stress component based on [40]. Next, participants performed the Cogstate tasks (follow-up assessment) and filled out the STAI and VAS. Side effects were then assessed, and the participants were debriefed. As part of this post-manipulation evaluation, the participants also estimated whether they were assigned to the active or sham stimulation condition; ≥80% of the participants believed they received active tDCS (range: 80.0–88.2%), with no significant difference between the two stimulation conditions (*χ*^2^ = 0.92, *p* = 0.821).

## 3. Results

Baseline differences were analyzed using analyses of variance (ANOVAs), with *stimulation* and *stress* as between-subject factors and *χ*^2^ analyses (for parametric and nonparametric variables, respectively). These analyses revealed no significant differences, as shown in Table 1.

Difference scores (Δ) were calculated for each outcome measure (follow-up—baseline) and analyzed using ANOVAs, with *stimulation* and *stress* as between-subject factors. Pearson product-moment correlation between the two Cogstate outcome measures was 0.19. Considering the weak correlation [41], an ANOVA rather than a MANOVA was deemed appropriate for analyzing the cognitive outcome measures. The analyses revealed a significant *stress* main effect on subjective measures of stress and anxiety; exposure to stress (TSST) increased participants’ anxiety and perceived stress (STAI-S: *p* = 0.001; VAS: *p* = 0.009). Regarding cognition, a significant *stimulation* × *stress* interaction was evident when analyzing performance in the OCL task (*p* = 0.047). Post hoc paired-sample *t*-tests (baseline vs. follow-up) performed separately on each group indicated enhanced post-manipulation performance among participants exposed to both sham tDCS and stress (TSST), *t*(33) = 4.16, *p* < 0.001. Pre-post differences in performance were not significant in the other three experimental groups (*p*s range: 0.249–0.940). No other significant main effects or interactions were found, including those performed on the *2*-Back task’s outcome measure (accuracy). Descriptive statistics and analyses of stress indicators and performance in Cogstate tasks are presented in Table 2.

## 4. Discussion

The current findings suggest that tDCS stimulation and social stress do not interact to influence visual working memory performance, at least as operationalized using the Cogstate *2*-Back task. Thus, the non-significant post hoc analyses in Ankri, Braw [13] did not reflect low power due to inadequate sample size; i.e., the sample size increased from 17 to 19 participants in Ankri, Braw [13] to 30–34 participants per experimental group in the current study. Findings also suggest that the other possible confounders mentioned earlier (i.e., use of cognitive tasks with a potential ceiling effect, lack of pre-manipulation cognitive testing, and use of a modified stress manipulation) had a negligible effect. This conclusion agrees with a recent study that used anodal offline stimulation of the left dlPFC [12]. In contrast, Bogdanov and Schwabe [10] found that stress impaired working memory performance, while tDCS stimulation mitigated this effect. However, the interaction only significantly impacted performance in the Corsi block task, while the analysis of the second task (i.e., digit span backwards) was only marginally significant [10]. Methodological differences between the studies (e.g., gender distribution, online vs. offline assessment of cognition, examiner-administered cognitive tasks vs. computerized testing, etc.) challenge comparisons between the studies. A conservative interpretation, however, is that right-prefrontal tDCS effects on working memory performance seem modest at best, and that the potential of tDCS to modify the impact of stress is limited.

The interaction between active tDCS and stress (TSST paradigm) significantly impacted participants’ performance on the second Cogstate task (OCL). Interestingly, OCL performance improved after exposure to stress, an effect that was absent when participants were exposed to both stress and tDCS stimulation. These findings bear some similarities to those of Ankri, Braw [13], though the cognitive domain assessed in the current study differed. More specifically, the Cogstate’s OCL task assesses immediate visual memory and learning. It is also less demanding on PFC resources than working memory tasks, which focus on accessing multimodal-memory neural networks modified by executive attention PFC-based mechanisms [15]. Note also that the significant interaction between stress and tDCS stimulation was not supported by post hoc analyses in Ankri, Braw [13]. Overall, our findings suggest that stress enhances immediate visual-memory learning, while right dlPFC stimulation may negate this effect. Such effects on working memory, at present, were not supported.

A dual outcome model of the interactive effect of right dlPFC stimulation and stress may be proposed. According to such a conceptualization, acute social stress could improve cognitive functioning due to its arousing properties [42], while more intense stress can overwhelm PFC-governed top-down regulatory processes and thereby impair performance [43,44]. The resulting performance may take on the form of an inverted-U pattern [45], which may explain the differential effects of tDCS that were evident in the current study. More specifically, applying right PFC anodal tDCS after being subjected to mild stress may temporarily suppress stress-induced cognitive enhancement, perhaps by strengthening executive right PFC regulatory network excitability (i.e., dampening the stress response) [11]. The PFC is involved in regulating HPA axis activity and, consequently, the individual’s stress response see [11]. The PFC in vitally involved in suppressing limbic circuit hyperactivity during stress [46,47]. Specifically, the right medial PFC (mPFC) has a pivotal role in the attenuation of stress responses by suppressing excitability in limbic regions, such as the insula, cingulate cortex, and networks within the right hippocampal gyrus [11]. This involvement impacts the secretion of cortisol, which affects regional cerebral blood flow and default-mode-network (DMN) activity, as well as modulates motor behavioral reactions to stress. Relatedly, cortisol hypersecretion in highly stressful situations can negatively affect synaptic connectivity in brain regions, such as the medial PFC, hippocampus, and amygdala [48]. This notable, as cortisol secretion affects on synaptic connectivity within prefrontal and limbic interconnected circuits, impacts brain regions that are essential for executive functions. (e.g., working memory). Correspondingly, in our previous study, which assessed the interactive effects of stress and right dlPFC tDCS on working memory performance, the group exposed to stress had significantly higher cortisol levels than the NoStress group [13]. This was associated with better working memory performance, indicating improved dlPFC functioning in the Stress group. However, the positive effects on working memory performance were abolished in the active tDCS group, implying that right dlPFC tDCS in young female participants may have interfered with the cortisol modulation of bilateral dlPFC connectivity, which is essential for WM performance. The cognitive-control neural networks during stress, which were described earlier, may also explain contrasting findings regarding the effects of stress. For example, a stress response—as evident in increased cortisol levels—was suggested to improve working memory performance in Lin, Leung [49], while it negatively impacted performance in other studies, e.g., [50]. PFC tDCS may help overcome the harmful effects of exposure to conditions associated with *severe* stress. This perspective may explain why Bogdanov and Schwabe [10] found that stress negatively impacted working memory performance while stimulation mitigated its impact, contrasting with the current study’s findings and the two earlier mentioned publications [12,13]. Correspondingly, higher subjective stress levels were reported in Bogdanov and Schwabe ( [10]; see Table 2 in p. 1432) compared to stress levels reported in the current study and in Ankri, Braw [13].

Several study limitations should be mentioned. First, the use of undergraduate female students restricts the generalizability of the findings. The recruitment of participants with more diverse baseline cognitive functioning is, therefore, advised in future studies. Second, using a single-blind design in the current study may have impacted the findings, and researchers are encouraged to use a double-blind design in future studies. Finally, the sample size in the current study was larger than those usually employed in similar studies. More specifically, an average of 14.6 participants per group were tested in previous studies using between-subjects designs [27]. The current study’s sample size is, therefore, not a limitation in a strict sense. However, researchers should still strive for similar if not larger samples, considering the lack of significant behavioral effects in at least some earlier studies, e.g., [11], and the general impression that any effect is likely to be small. The use of more cognitively taxing tasks, or other means to enhance learning, e.g., inducing mental fatigue; [51], is also encouraged and will increase the ability to detect significant changes in cognition following manipulations. In addition to the suggestions listed above, researchers may explore the sources of differences in findings between studies, including site of stimulation (right vs. left), gender, online vs. offline stimulation, etc. The use of stress manipulations differing in intensity is of particular importance and should preferably be incorporated into the same research design. Finally, the current study’s findings suggest a dissociation between the subjective experience of stress and cognitive functioning. More specifically, the stress manipulation led to an overall increase in reported stress levels and anxiety (VAS and STAI, respectively). This corresponds to previous findings, e.g., [10,11,50], and to the widespread support that the TSST manipulation has received over the years [38,52]. Therefore, systematic investigations on the impact of stress on subjective reports of anxiety and stress are warranted, as well as evaluating its direct impact on cognitive functioning.

To conclude, the study’s findings indicate that moderate social stress levels might temporarily enhance visual memory and learning, while right dlPFC tDCS suppresses this effect. In women, excitatory tDCS over the right dlPFC may indirectly disrupt stress-induced responses related to elevated cerebral blood flow (CBF) in the left hemisphere and reduced CBF to the dorsal ACC and left thalamus [53]. Thus, altering prefrontal network asymmetry prior to or during stress-induced conditions may interfere with top-down control of the limbic circuitry, which could also compromise visual learning. In contrast, visual working memory was not affected by stress, tDCS stimulation, or the interaction between the two. These findings support our hypothesis that moderate stress may excite certain hippocampal networks involved in memory formation due to a temporary stress-induced response of the HPA axis [54]. However, in women, it seems that altering prefrontal excitatory asymmetry may interfere with prefrontal-hippocampal stress responses that facilitate visual learning during moderate elevated levels of stress. Since several limitations noted in our previous study [13] were amended, we can be more confident in concluding that when using a similar experimental paradigm (i.e., right PFC anodal tDCS with same-sized cathodal reference, female participants, a social-stress manipulation that follows stimulation, and offline cognitive testing), only modest effects van be expected. We speculated that stress severity may be a critical factor in explaining inconsistencies between studies, considering its differential impact on limbic circuitry activations. Further research is needed to assess the effects of stress and the various factors (e.g., intensity of stress and prefrontal connectivity) leading to inconsistent findings in contemporary studies. Importantly, changes in PFC inhibitory neurotransmission are linked to stress and depression. Thus, in the clinical setting, focusing on dlPFC tDCS treatments that can improve intra-cortical GABA inhibition within PFC networks could enhance top-down control of anxiety-like behaviors in people diagnosed with major depressive disorder [46]. Finally, the current study’s findings suggest that levels of stress should be taken into consideration in the development of tDCS treatment protocols targeting the dlPFC in females experiencing acute stress. In reference to stressogenic psychiatric disorders, we suggest that monitoring stress may be particularly important in enhancing prefrontal tDCS treatment effects in individuals diagnosed with depression and post-traumatic stress disorder.

## Figures and Tables

**Figure 1 brainsci-13-01642-f001:**
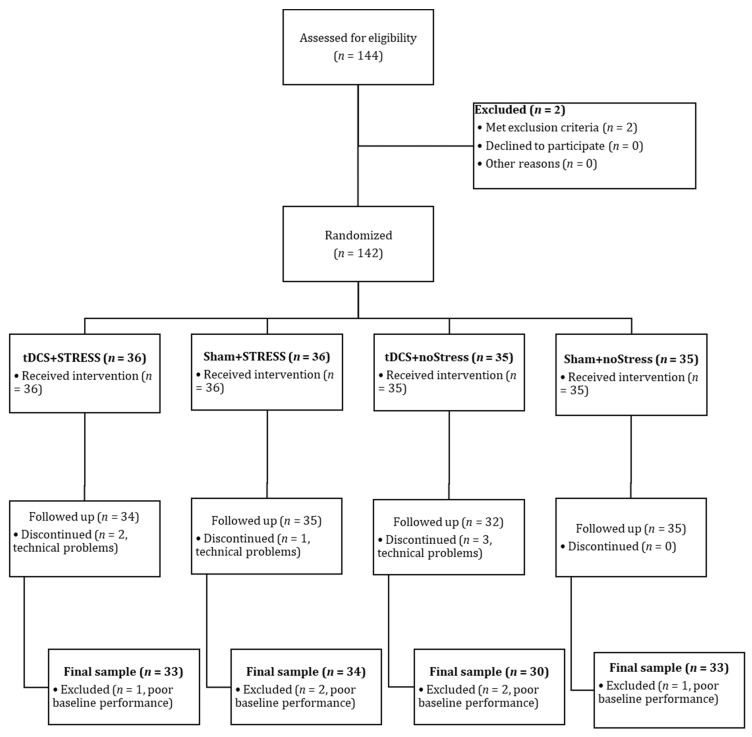
CONSORT flow diagram: number of participants included in enrollment, randomized allocation, follow-up, and analysis stages. Notes: Abbreviations: CONSORT: Consolidated standards of reporting trials; tDCS: Transcranial Direct Current Stimulation.

**Table 1 brainsci-13-01642-t001:** Comparison of baseline data per experimental group; STRESS + tDCS (*n* = 33), STRESS + Sham (*n* = 34), NoStress + tDCS (*n* = 30), and NoStress + Sham (*n* = 33).

Variables	STRESS		NoStress		
	tDCS	Sham	tDCS	Sham	Test of Difference †
Age	22.7 ± 1.7	22.3 ± 1.5	22.9 ± 2.1	22.4 ± 1.6	*F*(3,126) = 0.85, *p* = 0.467
Smoking	9, 28.1%	11, 32.4%	10, 33.3%	5, 15.2%	*χ*^2^ = 3.46, *p* = 0.325
Use of hormonal contraceptives	6, 18.2%	12, 35.3%	8, 27.6%	10, 30.3%	*χ*^2^ = 2.57, *p* = 0.463
STAI-T score	34.9 ± 8.9	34.6 ± 8.1	35.6 ± 9.8	34.4 ± 7.6	*F*(3,126) = 0.11, *p* = 0.955
STAI-S score	32.5 ± 7.0	33.2 ± 6.7	31.3 ± 6.6	31.7 ± 6.8	*F*(3,126) = 0.49, *p* = 0.683
VAS score	19.7 ± 25.6	19.5 ± 15.2	19.1 ± 26.3	15.2 ± 23.1	*F*(3,125) = 0.29, *p* = 0.835
Cogstate *2*-back accuracy ‡	123.9 ± 13.2	123.1 ± 14.5	126.1 ± 12.1	123.9 ± 13.2	*F*(3,126) = 0.49, *p* = 0.685
Cogstate OCL accuracy ‡	99.6 ± 7.5	101.8 ± 11.8	101.7 ± 8.5	103.5 ± 9.0	*F*(3,126) = 0.96, *p* = 0.413

Notes: † Analyses of variance (ANOVAs) were performed using stimulation (tDCS vs. sham) and stress manipulation (STRESS vs. noStress) as between-subject factors. ‡ Arcsine transformation of the square root of the proportion of correct response. All data are presented as *M* ± *SD*, except smoking and use of hormonal contraceptives (*n*, %). Abbreviations: *M*: mean; *SD*: standard deviation; STAI-T/S: State-Trait Anxiety Index- Trait/State; tDCS: Transcranial Direct Current Stimulation; VAS: visual analog scale.

**Table 2 brainsci-13-01642-t002:** Descriptive statistics and analyses of stress subjective indicators and Cogstate performance; STRESS + tDCS (*n* = 33), STRESS + Sham (*n* = 34), NoStress + tDCS (*n* = 30), and NoStress + Sham (*n* = 33).

Variable	Group	STRESS	NoStress			
		Δ (M ± SD)	Δ (M ± SD)	Stimulation	Stress	Stimulation × Stress
STAI-S total score (no.)	tDCS	4.5 ± 8.9	0.4 ± 6.00	*F*(1,126) = 0.55, *p* = 0.457	*F*(1,126) = 11.32, *p* = 0.001	*F*(1,126) = 0.10, *p* = 0.750
	Sham	3.9 ± 8.8	−1.0 ± 6.2			
VAS total score (no.)	tDCS	6.8 ± 29.8	−2.9 ± 25.7	*F*(1,124) = 0.36, *p* = 0.548	*F*(1,124) = 7.06, *p* = 0.009	*F*(1,124) = 0.39, *p* = 0.529
	Sham	12.6 ± 28.7	−3.00 ± 22.6			
*2*-back accuracy †	tDCS	7.3 ± 16.9	1.9 ± 12.7	*F*(1,126) = 0.66, *p* = 0.417	*F*(1,126) = 3.47, *p* = 0.065	*F*(1,126) = 0.014, *p* = 0.906
	Sham	9.2 ± 17.3	4.4 ± 13.8			
OCL accuracy †	tDCS	0.2 ± 10.7	0.9 ± 9.5	*F*(1,126) = 2.38, *p* = 0.126	*F*(1,126) = 2.59, *p* = 0.110	*F*(1,126) = 4.02, *p* = 0.047
	Sham	6.4 ± 9.1	0.1 ± 10.9			

Notes: † Arcsine transformation of the square root of the proportion of correct response. Data were multiplied (×100) for ease of presentation. Abbreviations: *M*: mean; *SD*: standard deviation; STAI-S: State–Trait Anxiety Inventory-State; tDCS: Transcranial Direct Current Stimulation; VAS: visual analog scale.

## Data Availability

The data that support the findings of this study are openly available in ZENODO at https://zenodo.org/search?page=1&size=20&q=4555163, reference number 4555163 (accessed on 3 November 2023).

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
