# Peer review of "Stress and Right Prefrontal Transcranial Direct Current Stimulation (tDCS) Interactive Effects on Visual Working Memory and Learning"

_brainsci, 2023, doi:10.3390/brainsci13121642_

Round 1

Reviewer 1 Report

Comments and Suggestions for Authors

The purpose of the study was to determine if there was an interaction between social stress and right dorsolateral prefrontal cortex (DLPFC) transcranial direct current application on working memory performance in young females. A total of 130 females were randomized into 4 groups (right DLPFC tDCS, SHAM, social stress, and control). The study was designed to be an extension of a previous study, but with more participants and a few other methodological improvements. That study found a tDCS by stress interaction for declarative memory. Accordingly, participants received tDCS of the right DLPFC for 20 minutes at 2 mA and the stress manipulations. Before and after stimulation, the n-Back and OCL tasks of the Cogstate battery were performed.

 The main findings were that both tDCS and social stress enhanced OCL performance. Neither intervention had an effect on the n-Back task. Overall, the results suggest tDCS influences on working memory are not as great as previously thought.

  Overall, this was a good study with several strengths such as a good design that built on previous research, a large sample size, and it was well-written. The only major weakness was the single blinded design. Although most tDCS studies need to be double blinded nowadays, I do not think in this case it lead to any bias as essentially the results were negative. This limitation was acknowledged by the authors. Thus, I only have minor changes to recommend and in no particular order. 

  1. Line 54 “inclearing” use a different word or maybe this is a typo.
  2. Line 55, I would not use the term “we hoped”. Change this wording.
  3. Line 80, maybe too many spaces after “Therefore,”
  4. Line 86, don’t have the subscript and text at the bottom of the page, just write the same info in the paragraph.
  5. Figure 1 the text is too light and faded and looks smudged.
  6. Table 1 is good but way too crowded. Determine somewhat to present the same data differently and spread things out.
  7. Line 222, I don’t think the sample size is a limitation I think it is a strength.
  8. Line 277 error in bolding the word The and no space after the colon.
  9. Bibliography has some errors. Sometimes the first letter of each word in an article title is capitalized, sometimes not. For instance compare references 1 and 2. This happens throughout the bibliography.

Comments on the Quality of English Language

minor proofreading needed

Author Response

Reviewer 1:

Comments and Suggestions for Authors

The purpose of the study was to determine if there was an interaction between social stress and right dorsolateral prefrontal cortex (DLPFC) transcranial direct current application on working memory performance in young females. A total of 130 females were randomized into 4 groups (right DLPFC tDCS, SHAM, social stress, and control). The study was designed to be an extension of a previous study, but with more participants and a few other methodological improvements. That study found a tDCS by stress interaction for declarative memory. Accordingly, participants received tDCS of the right DLPFC for 20 minutes at 2 mA and the stress manipulations. Before and after stimulation, the n-Back and OCL tasks of the Cogstate battery were performed.

 The main findings were that both tDCS and social stress enhanced OCL performance. Neither intervention had an effect on the n-Back task. Overall, the results suggest tDCS influences on working memory are not as great as previously thought.

  Overall, this was a good study with several strengths such as a good design that built on previous research, a large sample size, and it was well-written. The only major weakness was the single blinded design. Although most tDCS studies need to be double blinded nowadays, I do not think in this case it lead to any bias as essentially the results were negative. This limitation was acknowledged by the authors. Thus, I only have minor changes to recommend and in no particular order. 

Reply:

We thank the referee for taking the time to review the manuscript. We carefully read the comments and believe that the revised manuscript addressed all comments made by the referees, which were very helpful in improving the clarity and cohesiveness of the manuscript. As requested, we addressed methodological issues and amended the interpretation of the findings. Note that all changes in the manuscript were highlighted using a yellow background.

  1. Line 54 "inclearing" use a different word or maybe this is a typo.

Reply:

We thank you for noting the typo. The word was revised to "increasing" (see line 66 in the revised manuscript).

  1. Line 55, I would not use the term "we hoped". Change this wording.

Reply:

We agree that the original sentence was not phrased optimally. Moreover, it repeated our hypotheses (see last paragraph, p. 2 in the revised manuscript). It was, therefore, deleted from the revised manuscript. See 2nd paragraph, p. 2.

  1. Line 80, maybe too many spaces after "Therefore,"

Reply:

Done. See line 94 in the revised manuscript.

  1. Line 86, don't have the subscript and text at the bottom of the page, just write the same info in the paragraph.

Reply:

Done. The sentence was integrated into the paragraph. See p. 3. (participants subsection).

  1. Figure 1 the text is too light and faded and looks smudged.

Reply:

Done. The quality of Figure 1 was enhanced. Please see p. 3 in the revised manuscript.

  1. Table 1 is good but way too crowded. Determine somewhat to present the same data differently and spread things out.

Reply:

We appreciate the comment and agree that the original table was crowded. It was revised by: (a) Deleting the separate column for p values and integrating the data in the column detailing the statistical analyses. (b) Deleting the leftward column (titled 'Domain') and integrating data in the column detailing the variables. (b) Moving unnecessary data from the Table to the heading (i.e., group sizes) or notes (explaining regarding the data presented in the Table).

The table at, its revised form, is easier to follow, and we believe it conveys the data in a straightforward manner. Please note also that notes were added to both Table 1 and Table 2 for better clarity. They are presented in the revised manuscript, pp. 6–8.

  1. Line 222, I don't think the sample size is a limitation I think it is a strength.

Reply:

Our aim was aimed to increase the groups relative to our previous study; i.e., the sample size increased from 17–19 participants in Ankri, Braw [1]  to 30–34 participants per experimental group in the current study. We also aimed to increase the sample size relative to what is customary in the field; the average sample size is 14.6 participants for between-subjects designs and 17.9 for within-subject designs in brain stimulation studies.[2]. We agree that the sample size is a relative strength of the study. We, therefore, revised the limitations paragraph. The issue is presented at the end of the list of limitations and, importantly, phrased to reflect better the impact of sample size on the current study and similar studies in the field.

               Revised manuscript: "Finally, the sample size in the current study was larger than those usually employed in similar studies. More specifically, an average of 14.6 participants per group were tested in previous studies using between-subjects designs  [2]. The current study's sample size is, therefore, not a limitation in a strict sense. However, researchers should still strive for similar if not larger samples considering the lack of significant behavioral effects in at least some earlier studies [e.g., 3], and the general impression that any effects are likely to be small." (Discussion; lines 307–318). Note also that the issue was noted in other parts of the manuscript, including the Introduction ("Importantly, we incorporated insights from our earlier investigation incorporated in the research design. First, we enhanced statistical power by increasing the sample size."; pp. 2, lines 64–66), and Method ("In the current study, we aimed to increase the groups relative to our previous study [i.e., the sample size increased from 17–19 participants in 1 to 30–34 participants per experimental group in the current study] and relative to what is customary in the field. Regarding the latter, the average sample size is 14.6 participants for between-subjects designs and 17.9 for within-subject designs in brain stimulation studies.[2] We also used G*POWER 3.1.9.2 [4, 5] to examine the number of participants needed to detect an effect if one exists. Based on Mancuso, Ilieva [6], 28 participants per condition were deemed adequate, even assuming a small effect size. Note, also, that due to gender differences in lateralization of prefrontal network activation during verbal working memory storage [7], we recruited only female participants to increase the homogeneity of the sample. This was expected to decrease variance in outcome measures further and increase statistical power."; p. 3, lines 109–120).

  1. Line 277 error in bolding the word "The" and no space after the colon.

Reply:

Done. See line 392 in the revised manuscript.

  1. Bibliography has some errors. Sometimes the first letter of each word in an article title is capitalized, sometimes not. For instance compare references 1 and 2. This happens throughout the bibliography.

Reply:

 Done. The bibliography was revised according to the journal's instructions for authors.

Comments on the Quality of English Language: minor proofreading needed

Reply:

Following the comment, the paper was thoroughly proofread. We believe that all minor phrasing errors and typos were corrected and that the manuscript in its revised form is better streamlined and more straightforward to comprehend. For examples of revisions, see (a) Abstract: lines 16–26, 20–21, and 24–26. (b) Introduction: lines 33–34, 39–40, 53, 63, 65–66, 71, 75–79, 83, and 94–95. (c) Method: lines  151, 172, 173, 180, 185, and 186. (d) Results: line 208. (de Discussion: lines 222, 224, 225–227, 230, 233–234, 240–241, 242–247, 253, 259, 261–264, 287–289, 293–297, 239–331.

Submission Date

31 October 2023

Date of this review

10 Nov 2023 03:14:13

References:

  1. Ankri, Y.L.E., et al., The effects of stress and transcranial direct current stimulation (tdcs) on working memory: A randomized controlled trial. Cogn Affect Behav Neurosci, 2020. 20(1): p. 103-114.
  2. Medina, J. and S. Cason, No evidential value in samples of transcranial direct current stimulation (tdcs) studies of cognition and working memory in healthy populations. Cortex, 2017. 94: p. 131-141.
  3. Antal, A., et al., Transcranial electrical stimulation modifies the neuronal response to psychosocial stress exposure. Hum Brain Mapp, 2014. 35(8): p. 3750-9.
  4. Faul, F., et al., G*power 3: A flexible statistical power analysis program for the social, behavioral, and biomedical sciences. Behav Res Methods, 2007. 39(2): p. 175-91.
  5. Faul, F., et al., Statistical power analyses using g*power 3.1: Tests for correlation and regression analyses. Behav Res Methods, 2009. 41(4): p. 1149-60.
  6. Mancuso, L.E., et al., Does transcranial direct current stimulation improve healthy working memory?: A meta-analytic review. J Cogn Neurosci, 2016. 28(8): p. 1063-89.
  7. Meiron, O. and M. Lavidor, Unilateral prefrontal direct current stimulation effects are modulated by working memory load and gender. Brain Stimul, 2013. 6(3): p. 440-7.

Reviewer 2 Report

Comments and Suggestions for Authors

The authors hypothesized that right-prefrontal excitatory stimulation and stress would differentially impact visual learning. In addition, the authors intended to show that working memory performance would be unaffected due to its pronounced reliance on bilateral prefrontal function. The manuscript is very interesting and well structured. Furthermore, I believe that the topic covered is highly topical and has potential impact for the scientific community. However, I believe that the manuscript needs some modifications before it can be considered for publication. Below I give you my suggestions.

The introductory paragraph just highlights the role of the DLPFC. It would be appropriate to specify the importance and role of this area a little better. In this regard, I suggest to the authors two recent works in which the DLPFC was stimulated:

- Moscatelli et al., Acute non invasive brain stimulation improves performances in volleyball players. Physiol Behav. 2023 Nov 1;271:114356. doi: 10.1016/j.physbeh.2023.114356;

Figure 1 must necessarily be remodeled. the resolution is too low and the image is blurry. it is difficult to read the content.

The captions of the figures and tables should be rewritten. In particular, more detailed information should be provided in the captions in order to help readers understand the figures and tables well. Furthermore, all acronyms should be specified.

The conclusions are very similar to what is written at the beginning of the paragraph "Discussions". This part should be rewritten trying to provide conclusions on the study and provide possible future implications and possible practical applications.

Author Response

Reviewer 2:

Comments and Suggestions for Authors

The authors hypothesized that right-prefrontal excitatory stimulation and stress would differentially impact visual learning. In addition, the authors intended to show that working memory performance would be unaffected due to its pronounced reliance on bilateral prefrontal function. The manuscript is very interesting and well structured. Furthermore, I believe that the topic covered is highly topical and has potential impact for the scientific community. However, I believe that the manuscript needs some modifications before it can be considered for publication. Below I give you my suggestions.

Reply:

We thank the referee for taking the time to review the manuscript. We carefully read the comments and believe that the revised manuscript has been significantly improved as a result of the review process and that all comments have been adequately addressed. As suggested by the referee, we inserted amendments related to methodological issues and to the interpretation of the findings. Note that all changes in the manuscript were highlighted using a grey background.

The introductory paragraph just highlights the role of the DLPFC. It would be appropriate to specify the importance and role of this area a little better. In this regard, I suggest to the authors two recent works in which the DLPFC was stimulated:

- Moscatelli et al., Acute non invasive brain stimulation improves performances in volleyball players. Physiol Behav. 2023 Nov 1;271:114356. doi: 10.1016/j.physbeh.2023.114356;

 Reply:

Done. Following your comment, we detailed the contribution of the dlPFC in the revised Introduction section: "The dlPFC was chosen as it represents a critical anterior frontal neural network hub that is essential for facilitating immediate access to target stimuli representations and the mediation of action contingencies underlying the temporal organization of behaviors. More so, selective attention processes and retention of relevant visual-spatial working memory items can be improved by employing excitatory repetitive TMS (rTMS) over the left dlPFC, suggesting that dlPFC rTMS could also improve athletic visual-motor performance, which usually demands fast visual-spatial orienting-attention responses in dynamic "team-sport" environments [1]." (pp. 1–2, lines 41–48).

Figure 1 must necessarily be remodeled. the resolution is too low and the image is blurry. it is difficult to read the content.

Reply:

Done. Figure 1 was reconstructed and is currently adequately presented. See pp. 4–5 in the revised manuscript.

The captions of the figures and tables should be rewritten. In particular, more detailed information should be provided in the captions in order to help readers understand the figures and tables well. Furthermore, all acronyms should be specified.

Reply:

Done. The captions of Figure 1, Table 1, and Table 2 were revised. In addition, notes were added that provide additional information regarding measures (e.g., Cogstate outcome measures) and details the abbreviations that were used in the figure/table. See revised manuscript pp. 4, 6, and 8.

The conclusions are very similar to what is written at the beginning of the paragraph "Discussions". This part should be rewritten trying to provide conclusions on the study and provide possible future implications and possible practical applications.

Reply:

Done. We agree that the paragraph necessitated a revision, and it was reviewed and revised following your comment:

" To conclude, the study's findings indicate that moderate social stress levels might temporarily enhance visual memory and learning, while right dlPFC tDCS suppresses this effect. In women, excitatory tDCS over the right dlPFC may indirectly disrupt stress-induced responses related to elevated cerebral blood flow (CBF) in the left hemisphere and reduced CBF to the dorsal ACC and left thalamus Wang et al., 2007). Thus, altering prefrontal network asymmetry prior or during stress-induced conditions may interfere with top-down control of limbic circuitry, which could also compromise visual learning. In contrast, visual working memory was not affected by stress, tDCS stimulation, or the interaction between the two. These findings support our hypothesis that moderate stress may excite certain hippocampal networks involved in memory formation due to a temporary stress-induced response in the HPA axis [2]. However, in women, it seems that altering prefrontal excitatory asymmetry may interfere with prefrontal-hippocampal stress responses that facilitate visual learning during moderate elevated levels of stress. Since several limitations noted in our previous study [3] were amended, we can be more confident in concluding that when using a similar experimental paradigm (i.e., right PFC anodal tDCS with same-sized cathodal reference, female participants, a social-stress manipulation that follows stimulation, and offline cognitive testing) only modest effects are expected. We speculated that stress severity may be a critical factor in explaining inconsistencies between studies, considering its differential impact on limbic circuitry activations. Further research is needed to assess the effects of stress and the various factors (e.g., intensity of stress, prefrontal connectivity) leading to inconsistent findings in contemporary studies. Importantly, changes in PFC inhibitory neurotransmission are linked to stress and depression. Thus, in the clinical setting, focusing on dlPFC tDCS treatments that can improve intra-cortical GABA inhibition within PFC networks could enhance top-down control of anxiety-like behaviors in people diagnosed with major depressive disorder [4]. Finally, the current study’s findings suggest that levels of stress should be taken into consideration in the development of tDCS treatment protocols targeting the dlPFC in females experiencing acute stress. In relevance to stressogenic psychiatric disorders, we suggest that monitoring stress may be particulty important in enhancing prefrontal tDCS-treatment procogntive effects of individulas diagnosed with depression and post-traumatic-stress disorder." [revisions marked using a gray background; changes made upon the request of the first reviewer were marked using a yellow background]. See pp. 10–11 in the revised manuscript.

Submission Date

31 October 2023

Date of this review

04 Nov 2023 16:18:09

References:

  1. Moscatelli, F., et al., Acute non invasive brain stimulation improves performances in volleyball players. Physiol Behav, 2023. 271: p. 114356.
  2. McEwen, B.S. and P.J. Gianaros, Central role of the brain in stress and adaptation: Links to socioeconomic status, health, and disease. Ann N Y Acad Sci, 2010. 1186: p. 190-222.
  3. Ankri, Y.L.E., et al., The effects of stress and transcranial direct current stimulation (tdcs) on working memory: A randomized controlled trial. Cogn Affect Behav Neurosci, 2020. 20(1): p. 103-114.
  4. Ghosal, S., B. Hare, and R.S. Duman, Prefrontal cortex gabaergic deficits and circuit dysfunction in the pathophysiology and treatment of chronic stress and depression. Curr Opin Behav Sci, 2017. 14: p. 1-8.

Reviewer 3 Report

Comments and Suggestions for Authors

The manuscript describes a very interesting study about tDCS and WM. The introduction is well written and the authors described the previous studies, highlighting the pros and cons of the technique. DlPFC is a crucial brain area for WM and several studies have been performed. Indeed, genetic fMRI studies showed how relevant is in WM in schizophrenic patients explain the heterogeneity in the functional behavior of dlPFC. Similarly, despite I know the reason of the choice about female cohort of participants, this should be stated in a better way. I’ve appreciated this and I agree with the authors because the female brain is still not well studied and understood. In this way, I advise to delete “ Due to gender differences in lateralization of prefrontal network activation durining verbal working memory storage [26], we recruited only female participants to increase homogeneity of sample.” This info should be integrated in the main text avoiding the redundancy.

The methods section is well explained allowing the replication of the study. However, I advise the authors to add the estimation of the sample size and in which manner they made it. I agree with the timeline of the study and test used. It is very rigorous.

About the data analysis, I suppose that a MANOVA should be better, but the ANOVA can be acceptable.

The discussion is very interesting and is line with the results. The dual model is also well described ad supported by the performed experiment. However, about STRESS a more rigorous terminology (technical) is needed. I’d like to read something about the role of cortisol, stress and PFC, but it is only an advice.

Author Response

Reviewer 3:

Comments and Suggestions for Authors

The manuscript describes a very interesting study about tDCS and WM. The Introduction is well written and the authors described the previous studies, highlighting the pros and cons of the technique. DlPFC is a crucial brain area for WM and several studies have been performed. Indeed, genetic fMRI studies showed how relevant is in WM in schizophrenic patients explain the heterogeneity in the functional behavior of dlPFC. Similarly, despite I know the reason of the choice about female cohort of participants, this should be stated in a better way. I've appreciated this and I agree with the authors because the female brain is still not well studied and understood. In this way, I advise to delete "Due to gender differences in lateralization of prefrontal network activation durining verbal working memory storage [26], we recruited only female participants to increase homogeneity of sample." This info should be integrated in the main text avoiding the redundancy.

Reply:

We thank the referee for taking the time to review the manuscript. We carefully read the comments and believe that the revised manuscript addressed all comments made by the referee which were very helpful in improving the clarity and cohesiveness of the manuscript.  As requested, we addressed all methodological issues and made amendments to the interpretation of the findings. Note that all changes in the manuscript were noted using a green background.

The methods section is well explained allowing the replication of the study. However, I advise the authors to add the estimation of the sample size and in which manner they made it. I agree with the timeline of the study and test used. It is very rigorous.

Reply:

Done. We appreciate your comment regarding the timeline, and we added an estimation of the sample size to the revised manuscript following your comment.

Revised manuscript (p. 3): "In the current study, we aimed to increase the groups relative to our previous study [i.e., the sample size increased from 17–19 participants in 1 to 30–34 participants per experimental group in the current study] and relative to what is customary in the field. Regarding the latter, the average sample size is 14.6 participants for between-subjects designs and 17.9 for within-subject designs in brain stimulation studies.[2] We also used G*POWER 3.1.9.2 [3, 4] to examine the number of participants needed to detect an effect if one exists. Based on Mancuso, Ilieva [5], 28 participants per condition were deemed adequate, even assuming a small effect size. Note, also, that due to gender differences in lateralization of prefrontal network activation during verbal working memory storage [6], we recruited only female participants to increase homogeneity of sample. This was expected to further decrease variance in outcome measures and increase statistical power."

About the data analysis, I suppose that a MANOVA should be better, but the ANOVA can be acceptable.

Reply:

Pearson product-moment correlation between the two Cogstate outcome measures was .19. Considering the weak correlation [7], an ANOVA rather than a MANOVA was deemed appropriate for analyzing the cognitive outcome measures. This was noted in the revised Results section (p. 7).

The discussion is very interesting and is line with the results. The dual model is also well described and supported by the performed experiment. However, about STRESS a more rigorous terminology (technical) is needed. I'd like to read something about the role of cortisol, stress and PFC, but it is only an advice.

Reply:

We appreciate the kind words regarding the Discussion and presentation of the dual model. We agree that a more through discussion information was needed about the effects of stress. We therefore added the following section to the revised Discussion section: "The PFC is involved in regulating HPA axis activity and, consequently, the individual’s stress response [see 8]. This involvement impacts the secretion of cortisol, which affects regional cerebral blood flow and default-mode-network (DMN) activity, and modulates motor behavioral-reactions to stress. Specifically, the right medial PFC (mPFC) has a pivotal role in the attenuation of stress responses by suppressing excitability in limbic regions such as the insula, cingulate cortex, and networks within the right hippocampal gyrus [8]. In highly stressful situations, cortisol hypersecretion can negatively affect synaptic connectivity in brain regions such as the medial PFC, hippocampus, and amygdala [9]. This latter suggestion corresponds to the vital involvement of PFC in suppressing limbic circuit hyperactivity during stress [10, 11]. Importantly, cortisol secretion can affect synaptic connectivity within prefrontal and limbic interconnected circuits, which are essential for executive functions such as working memory. Correspondingly, in our previous study, which assessed the interactive effects of stress and right dlPFC tDCS on working memory performance, the group exposed to stress had significantly higher cortisol levels versus the NoStress group [1]. This was associated with better working memory performance, indicating improved dlPFC functioning in Stress group. However, the positive effects on working memory performance were abolished in the active tDCS group, implying that right dlPFC tDCS in young female participants may have interfered with cortisol modulation of bilateral dlPFC connectivity, which is essential for WM performance. The cognitive-control neural networks during stress, which was described earlier,…" (p. 9, lines 264–284).

Submission Date

31 October 2023

Date of this review

09 Nov 2023 16:40:35

References:

  1. Ankri, Y.L.E., et al., The effects of stress and transcranial direct current stimulation (tdcs) on working memory: A randomized controlled trial. Cogn Affect Behav Neurosci, 2020. 20(1): p. 103-114.
  2. Medina, J. and S. Cason, No evidential value in samples of transcranial direct current stimulation (tdcs) studies of cognition and working memory in healthy populations. Cortex, 2017. 94: p. 131-141.
  3. Faul, F., et al., G*power 3: A flexible statistical power analysis program for the social, behavioral, and biomedical sciences. Behav Res Methods, 2007. 39(2): p. 175-91.
  4. Faul, F., et al., Statistical power analyses using g*power 3.1: Tests for correlation and regression analyses. Behav Res Methods, 2009. 41(4): p. 1149-60.
  5. Mancuso, L.E., et al., Does transcranial direct current stimulation improve healthy working memory?: A meta-analytic review. J Cogn Neurosci, 2016. 28(8): p. 1063-89.
  6. Meiron, O. and M. Lavidor, Unilateral prefrontal direct current stimulation effects are modulated by working memory load and gender. Brain Stimul, 2013. 6(3): p. 440-7.
  7. Cohen, J., A power primer. Psychol Bull, 1992. 112(1): p. 155-159.
  8. Antal, A., et al., Transcranial electrical stimulation modifies the neuronal response to psychosocial stress exposure. Hum Brain Mapp, 2014. 35(8): p. 3750-9.
  9. Duque, A., I. Cano-Lopez, and S. Puig-Perez, Effects of psychological stress and cortisol on decision making and modulating factors: A systematic review. Eur J Neurosci, 2022. 56(2): p. 3889-3920.
  10. Ghosal, S., B. Hare, and R.S. Duman, Prefrontal cortex gabaergic deficits and circuit dysfunction in the pathophysiology and treatment of chronic stress and depression. Curr Opin Behav Sci, 2017. 14: p. 1-8.
  11. Kalin, N.H., Prefrontal cortical and limbic circuit alterations in psychopathology. Am J Psychiatry, 2019. 176(12): p. 971-973.

Round 2

Reviewer 3 Report

Comments and Suggestions for Authors

Authors addressed all my issues, congratulation for this interesting study!